# High-Precision Dichotomous Image Segmentation via Probing Diffusion Capacity

**Qian Yu**[1,2*] **Peng-Tao Jiang**[2*†] **Hao Zhang**[2] **Jinwei Chen**[2] **Bo Li**[2] **Lihe Zhang**[1†] **Huchuan Lu**[1]
[1]Dalian University of Technology    [2]vivo Mobile Communication Co., Ltd
{ms.yuqian}@mail.dlut.edu.cn, {pt.jiang,libra}@vivo.com,
{zhanglihe,luhuchuan}@dlut.edu.cn

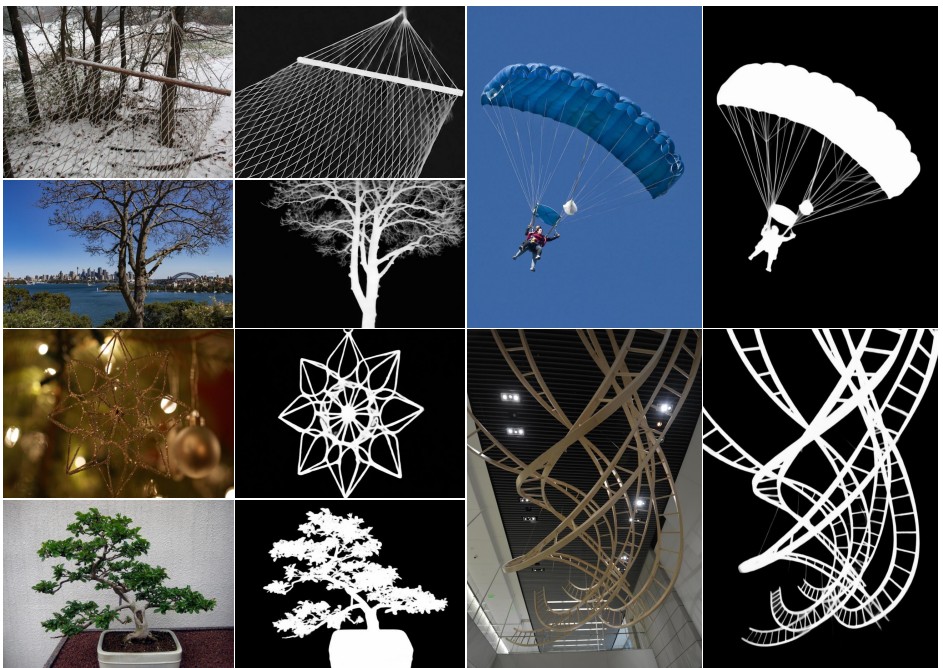

Figure 1: Dichotomous image segmentation results using our DiffDIS.

## Abstract

In the realm of high-resolution (HR), fine-grained image segmentation, the primary challenge is balancing broad contextual awareness with the precision required for detailed object delineation, capturing intricate details and the finest edges of objects. Diffusion models, trained on vast datasets comprising billions of image-text pairs, such as SD V2.1, have revolutionized text-to-image synthesis by delivering exceptional quality, fine detail resolution, and strong contextual awareness, making them an attractive solution for high-resolution image segmentation. To this end, we propose DiffDIS, a diffusion-driven segmentation model that taps into the potential of the pre-trained U-Net within diffusion models, specifically designed for high-resolution, fine-grained object segmentation. By leveraging the robust generalization capabilities and rich, versatile image representation prior of the SD models, coupled with a task-specific stable one-step denoising approach, we significantly reduce the inference time while preserving high-fidelity, detailed generation. Additionally, we introduce an auxiliary edge generation task to not only enhance the preservation of fine details of the object boundaries, but reconcile the probabilistic nature of diffusion with the deterministic demands of segmentation. With these refined strategies in place, DiffDIS serves as a rapid object mask generation model, specifically optimized for generating detailed binary maps at high resolutions, while demonstrating impressive accuracy and swift processing.

---

[*]Work was done during interning at vivo. The first two authors share equal contributions.
[†]Corresponding authors.

Experiments on the DIS5K dataset demonstrate the superiority of DiffDIS, achieving state-of-the-art results through a streamlined inference process. The source code will be publicly available at DiffDIS.

# 1 INTRODUCTION

High-accuracy dichotomous image segmentation (DIS) (Qin et al., 2022) aims to accurately identify category-agnostic foreground objects within natural scenes, which is fundamental for a wide range of scene understanding applications, including AR/VR applications (Tian et al., 2022; Qin et al., 2021), image editing (Goferman et al., 2011), and 3D shape reconstruction (Liu et al., 2021). Different from existing segmentation tasks, DIS focuses on challenging high-resolution (HR) fine-grained object segmentation, regardless of their characteristics. These objects often encompass a greater volume of information and exhibit richer detail, thereby demanding more refined feature selection and more sophisticated algorithms for segmentation. Current CNN-(Pei et al., 2023; Qin et al., 2022) and Transformer-based (Xie et al., 2022; Kim et al., 2022; Yu et al., 2024) methods, despite their robust feature extraction capabilities, often face challenges in balancing receptive field expansion with detail preservation in high-resolution images (Xie et al., 2022; Yu et al., 2024).

Diffusion probabilistic models (DPMs), by predicting noise variables across the entire image, have been demonstrated by numerous studies (Ji et al., 2023; Wang et al., 2023) to show promise in maintaining a global receptive field while more accurately learning the target distribution. Moreover, Stable Diffusion (SD) (Rombach et al., 2022) has emerged as a significant leap forward in the domain of DPMs. Trained on vast datasets comprising billions of images, it exhibits robust generalization capabilities and offers a rich, versatile image representation, positioning it as an ideal candidate for tasks demanding both macroscopic context and microscopic precision. Its power is further underscored by its significant contributions to fine-grained feature extraction, as evidenced by recent studies (Ke et al., 2024; Zhang et al., 2024; Zavadski et al., 2024; Hu et al., 2023; Xu et al., 2024) that have harnessed the SD's capabilities to capture the subtleties of detail. The advancements in these methods inspired us that diffusion could be a powerful tool for improving the accuracy and robustness of high-resolution image segmentation.

However, there are several challenges when leveraging the diffusion model to DIS: (1) The inherent high-dimensional, continuous feature space of DPM conflicts with the discrete nature of binary segmentation, potentially leading to a discrepancy in the predictive process. (2) Moreover, the diffusion models often suffer from lengthy inference time. Due to the recurrent nature of diffusion models, it usually takes more than 100 steps for DPM to generate satisfying results (Ke et al., 2024), which further exacerbates the already slow inference speeds of HR images due to their substantial data volume. (3) There is a fundamental conflict between the stochastic nature of diffusion and the deterministic outcomes required for image perception tasks.

Against this backdrop, we propose DiffDIS, addressing both aforementioned challenges and task-specific complexities of DIS, focusing on enhanced processing speed, stronger detail perception, and higher determinism, achieved through the

Table 1: The restorative capability of VAE.

| Metric | $F_\beta^{max} \uparrow$ | $E_\phi^m \uparrow$ | $S_m \uparrow$ | $\mathcal{M} \downarrow$ |
|---|---|---|---|---|
| VAE | 0.993 | 0.999 | 0.985 | 0.002 |

following strategies: First, we found that the Variational Autoencoder (VAE) (Kingma, 2013) has demonstrated the capability to nearly achieve perfect reconstruction of binary masks (See Tab. 1). Therefore, following SD (Rombach et al., 2022), by mapping the masks through the VAE into the latent space, we can not only effectively leverage the strengths of diffusion models for denoising and refinement within it, but also significantly reduces the computational cost associated with processing HR image segmentation. Then, by employing a direct one-step denoising paradigm (See Eqn. 3) and integrating the pretrained parameters of SD-Turbo (Sauer et al., 2023; Parmar et al., 2024), which feature a smoother probability curve compared to the standard SD V2.1 parameters, we streamline the network into an end-to-end model. This approach facilitates one-step denoising while leveraging its robust generalization capabilities and preserving the intricate details of high-resolution objects. Moreover, we introduce a joint predicting strategy for mask and edge, which enhances mask generation through finer constraints while improving controllability of diffusion models for perception tasks. Finally, we condition the diffusion model on RGB latent representations and propose Scale-

Wise Conditional Injection to enable multi-granular, long-range feature interactions, ensuring more refined feature preservation and selection.

Generally, our main contributions can be summarized as follows: (1) We propose DiffDIS, leveraging the powerful prior of diffusion models for the DIS task, elegantly navigating the traditional struggle to effectively balance the trade-off between receptive field expansion and detail preservation in traditional discriminative learning-based methods. (2) We transform the recurrent nature of diffusion models into an end-to-end framework by implementing straightforward one-step denoising, significantly accelerating the inference speed. (3) We introduce an auxiliary edge generation task, complemented by an effective interactive module, to achieve a nuanced balance in detail representation while also enhancing the determinism of the generated masks. (4) We advance the field forward by outperforming almost all metrics on the DIS benchmark dataset, and thus establish a new SoTA in this space. Additionally, it boasts an inference speed that is orders of magnitude faster than traditional multi-step diffusion approaches without compromising accuracy.

## 2 RELATED WORKS

### 2.1 CONVENTIONAL APPROACH TO DEAL WITH DIS WORKS

Dichotomous Image Segmentation (DIS) is formulated as a category-agnostic task that focuses on accurately segmenting objects with varying structural complexities, independent of their specific characteristics. It includes high-resolution images containing salient (Pang et al., 2020; Zhao et al., 2020; 2024c), camouflaged (Pang et al., 2024b; 2022; Fan et al., 2020; Pang et al., 2024a), and meticulous (Liew et al., 2021; Yang et al., 2020) instances in various backgrounds, integrating several context-dependent (Zhao et al., 2024b;a) segmentation tasks into a unified benchmark. When dealing with DIS, it is essential to consider the demand for highly precise object delineation, capturing the finest internal details of objects. Upon introducing the DIS dataset, Qin et al. also presented IS-Net (Qin et al., 2022), a model specifically crafted to address the DIS challenge, utilizing the $U^2$Net and intermediate supervision to mitigate overfitting risks. PF-DIS (Zhou et al., 2023) utilized a frequency prior generator and featured harmonization module to identify fine-grained object boundaries in DIS. UDUN (Pei et al., 2023) proposed a unite-divide-unite scheme to disentangle the trunk and structure segmentation for DIS. InSPyReNet (Kim et al., 2022) was constructed to generated HR outputs with a multi-resolution pyramid blending at the testing stage. Recent approaches have incorporated multi-granularity cues that harness both global and local information to enhance localization accuracy and detail fidelity. BiRefNet (Zheng et al., 2024) employed a bilateral reference strategy, leveraging patches of the original images at their native scales as internal references and harnessing gradient priors as external references. Recently, MVANet (Yu et al., 2024) modeled the DIS task as a multi-view object perception problem, leveraging the complementary localization and refinement among views to process HR, fine-detail images. Despite their commendable performance, existing methods have not effectively balanced the semantic dispersion of HR targets within a limited receptive field with the loss of high-precision details in a larger receptive field when tackling DIS. Compared to conventional approaches, our diffusion-based framework has excelled in achieving high-quality background removal, with fast processing times, and offers a straightforward integration.

### 2.2 DIFFUSION MODELS FOR DENSE PREDICTION AND EFFICIENT INFERENCE IN DIFFUSION

With the recent success of diffusion models in generation tasks, there has been a noticeable rise in interest in incorporating them into dense visual prediction tasks. Several pioneering works attempted to apply the diffusion model to visual perception tasks, *e.g.*image segmentation (Amit et al., 2021; Ji et al., 2023; Wang et al., 2023), matting (Hu et al., 2023; 2024), depth estimation (Ke et al., 2024; Zhang et al., 2024; Zavadski et al., 2024), edge detection (Ye et al., 2024) *et al.*. Since the pioneering work (Amit et al., 2021) introduced diffusion methods to solve image segmentation, several follow-ups use diffusion to attempt their respective tasks. (Ji et al., 2023) formulated the dense visual prediction tasks as a general conditional denoising process. (Hu et al., 2023) pioneered the use of diffusion in matting, decoupling encoder and decoder to stabilize performance with uniform time intervals. (Hu et al., 2024) ingeniously trained the model to paint on a fixed pure green screen backdrop. (Ye et al., 2024) utilized a decoupled architecture for faster denoising and an adaptive

Fourier filter to adjust latent features at specific frequencies. In the depth estimation field, several recent works have leveraged diffusion for high-fidelity, fine-grained generation. (Ke et al., 2024) introduced a latent diffusion model based on SD (Rombach et al., 2022), with fine-tuning for depth estimation, achieving strong performance on natural images. (Zavadski et al., 2024) extracts a rich, frozen image representation from SD, termed preimage, which is then refined for downstream tasks. Inspired by their work, we observe that diffusion-generative methods naturally excel at modeling complex data distributions and generating realistic texture details.

However, traditional multi-step generation models encounter a challenge: the recurrent structure of diffusion models often necessitates over 100 steps to produce satisfactory outputs. In response, various proposals have emerged to expedite the inference process. Distillation-based methods have demonstrated remarkable speedups by optimizing the original diffusion model's weights with enhanced schedulers or architectures. (Luo et al., 2023) achieves a few-step inference in conditional image generation through self-consistency enforcement. SD-Turbo (Sauer et al., 2023) employs Adversarial Diffusion Distillation (ADD), utilizing a pre-trained SD model to denoise images and calculate adversarial and distillation losses, facilitating rapid, high-quality generation. Recently, GenPercept (Xu et al., 2024) presents a novel perspective on the diffusion process as an interpolation between RGB images and perceptual targets, effectively harnessing the pre-trained U-Net for various downstream image understanding tasks. Given that SD-Turbo refines both high-frequency and low-frequency information through distillation (Sauer et al., 2023), by harnessing its efficient prior for image generation, we can sustain competitive performance in a single-step denoising scenario while achieving high-fidelity, fine-grained mask generation (See Fig. 1).

## 3 PRELIMINARIES

Diffusion probabilistic models (Ho et al., 2020) have emerged as highly successful approaches for generating images by modeling the inverse process of a diffusion process from Gaussian noise. It defines a Markovian chain of diffusion forward process $q(x_t|x_0)$ by gradually adding noise to input data $x_0$ :

$$x_t = \sqrt{\bar{\alpha}_t}x_0 + \sqrt{1 - \bar{\alpha}_t}\epsilon, \quad \epsilon \sim \mathcal{N}(0, I), \tag{1}$$

where $\epsilon$ is a pure Gaussian noise map. As $t$ increases, $\bar{\alpha}_t$ gradually decreases, leading $x_t$ to approximate the Gaussian noise. By predicting the noise, the loss can be written as:

$$\mathcal{L}(\epsilon_\theta) = \sum_{t=1}^{T} \mathbb{E}_{x_0 \sim q(x_0), \epsilon \sim \mathcal{N}(0,I)} \left[ \left\| \epsilon_\theta \left( \sqrt{\bar{\alpha}_t}x_0 + \sqrt{1 - \bar{\alpha}_t}\epsilon \right) - \epsilon \right\|_2^2 \right], \tag{2}$$

During the reverse process in DDPM (Ho et al., 2020), given a random sampled Gaussian noise $x_T \sim \mathcal{N}(0, I)$, we utilize a Gaussian distribution $p_\theta(\mathbf{x}_{t+1} \mid \mathbf{x}_t)$ to approximate the true posterior distribution, whose mean is estimated by the neural network, and the variance, denoted as $\sigma_t^2$, is derived from the noise schedule. To this end, we can repeat the following denoising process for $t \in \{T, T-1, \ldots, 1\}$ to predict final denoised result $x_0$.

$$x_{t-1} = \frac{1}{\sqrt{\alpha_t}} \left( x_t - \frac{1 - \alpha_t}{\sqrt{1 - \bar{\alpha}_t}} \epsilon_\theta (x_t, t) \right) + \sigma_t \epsilon, \tag{3}$$

In the context of our method, during the training phase, instead of randomly selecting $t$, we initialize $t$ to its maximum value $T$, *i.e.*, 999. This initialization enables the network to directly learn the probability distribution for a single-step denoising process. From Eqn. 3, the one-step denoising formula can be derived as follows. Since $\sigma_T$ is relatively small when $T = 999$, its impact on the $x_0$ can be considered negligible. The formula is given by:

$$\hat{x}_0 = \texttt{Denoise}\left(\epsilon_\theta(x_T, T, cond, d_{lab}), \epsilon\right) = \frac{x_T - \sqrt{1 - \bar{\alpha}_T}\epsilon_\theta(x_T, T)}{\sqrt{\bar{\alpha}_T}}, \tag{4}$$

where $cond$ represents the conditional input, specifically the image latent, while $d_{lab}$ denotes discriminative labels used to generate batch-discriminative embeddings (See Sec. 4.2 ) . Accordingly, the training objective of our method can be reformulated as minimizing the expected squared error between the denoised output and the ground truth latent $x_0$, as expressed by:

$$\min_\theta \mathbb{E}_{t, \epsilon, cond} \|\hat{x}_0 - x_0\|_2^2, \tag{5}$$

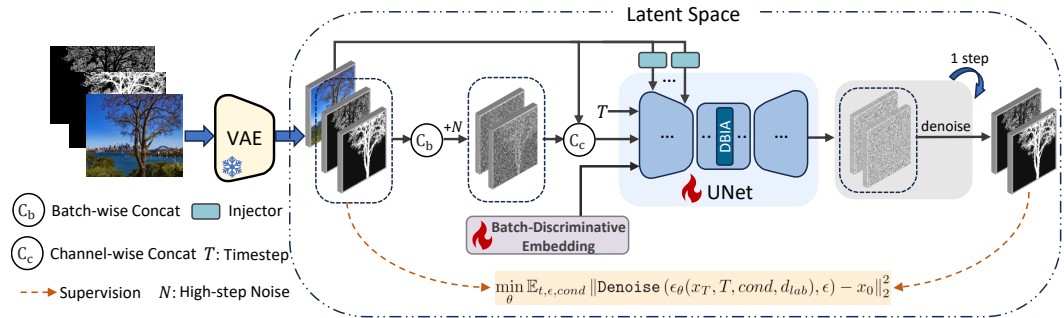

Figure 2: Overall framework of DiffDIS. To start with, the inputs are encoded into the latent space. We concatenate the noisy mask and its corresponding edge latent along the batch dimension, utilizing batch-discriminative embedding for differentiation. Then, we employ the RGB latent as a conditioning factor through channel-wise concatenation, along with multi-scale conditional injection into the U-Net encoder layers. The noise obtained is processed through a direct single-step denoising approach to yield a clean latent prediction.

# 4 METHOD

## 4.1 OVERALL ARCHITECTURE

The workflow of DiffDIS is outlined in Alg. 1: First, the inputs are encoded into the VAE's latent space, capturing the essential features in a continuous, high-dimensional representation. Then, a denoising model is applied to mitigate noise, aiming for the best denoising effects to refine the representations and enhance the clarity of the features. We utilize the RGB latent as a conditioning factor to assist in generating more authentic structural details for the mask. Meanwhile, we improve the information flow by multi-tasking to further boost the accuracy. We concatenate the mask and edge on a batch basis and apply Batch Discriminative Embedding (Fu et al., 2024) to obtain distinctive embeddings for each. These embeddings are then fed into the denoising network together to help distinguish between mask and edge.

**Algorithm 1** Training Process

**Input:** $cond$: conditional image latent, $m$: mask latent, $e$: edge latent, $d_{lab}$: discriminative labels
**while** not converged **do**
    $t = T$
    $\epsilon \sim \mathcal{N}(0, I)$
    $d_{emb} = \text{BDE}(d_{lab})$
    $m_t = \sqrt{\bar{\alpha}_t}m + \sqrt{1 - \bar{\alpha}_t}\epsilon$
    $e_t = \sqrt{\bar{\alpha}_t}e + \sqrt{1 - \bar{\alpha}_t}\epsilon$
    $\epsilon_{\text{pred}_m}, \epsilon_{\text{pred}_e} = \epsilon_\theta(m_t, e_t, cond, t, d_{emb})$
    $l_{\text{pred}_m} = \left(m_t - \sqrt{1 - \bar{\alpha}_t} \times \epsilon_{\text{pred}_m}\right) / \sqrt{\bar{\alpha}_t}$
    $l_{\text{pred}_e} = \left(e_t - \sqrt{1 - \bar{\alpha}_t} \times \epsilon_{\text{pred}_e}\right) / \sqrt{\bar{\alpha}_t}$
    Perform Gradient descent steps on $\nabla_\theta \mathcal{L}_{\text{total}}(\theta)$
**end while**
**return** $\theta$

Within the U-Net architecture, specifically in the mid-block where semantic information is most concentrated, we strategically integrate an additional attention module called Detail-Balancing Interactive Attention. This module not only enhances the determinism of the generative model's predictions (See Fig. 4) but also facilitates a concise yet potent interaction between the high-dimensional noisy features of masks and edges, as detailed in Section 4.2. Upon acquiring the noise estimated from the U-Net, we proceed with a direct one-step denoising transition from the high-step noisy latents, in accordance with the methodology outlined in Eqn. 4. We subsequently utilize the initial mask and edge latent codes obtained from the VAE for direct supervision of the denoised mask and edge latent codes. For an illustrative training diagram, refer to Fig. 2.

## 4.2 EDGE-ASSISTED TRAINING STRATEGY

In high-resolution tasks, the complexity and detail richness make it tough to capture fine features. Furthermore, for diffusion-based architectures, accurately depicting detailed features during few-step generation poses a challenge, especially as details may be overlooked during denoising in the latent space, which is often reduced to $\frac{1}{8}$ the size of the input. Some studies (Pei et al., 2023; Chen et al., 2023; Zhao et al., 2022) have indicated that incorporating additional edge constraints improves the boundary segmentation performance of masks. Given these considerations, we propose an integrated and streamlined network architecture that concurrently predicts noise for both the mask and edge, leveraging auxiliary edge information to constrain the network. These dual prediction streams operate within the same network structure and share parameters, with batch discriminative embed-

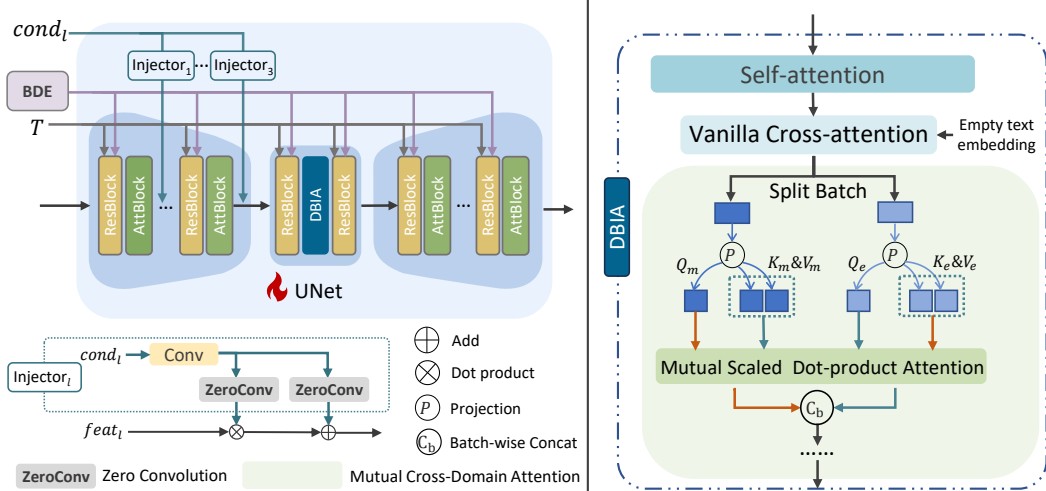

Figure 3: The structure of Batch-Discriminative Embedding (BDE) and the Detail-Balancing Interactive Attention (DBIA) .

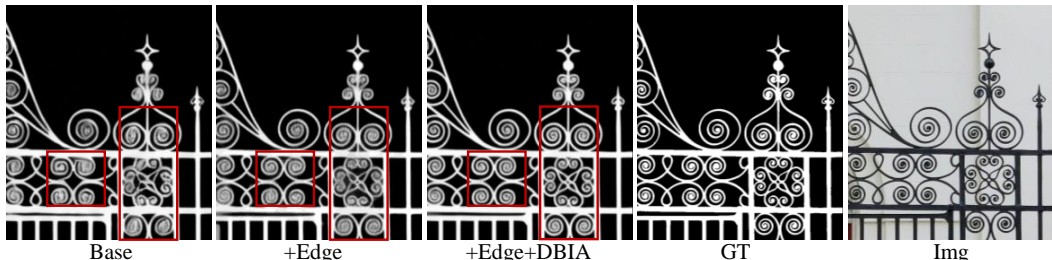

Base      +Edge      +Edge+DBIA      GT      Img

Figure 4: Visualizing the reduction of diffusion process stochasticity through edge integration. With the addition of the edge auxiliary prediction task, the controllability and alignment of the generated mask have been enhanced, particularly in the areas highlighted by red boxes. The introduction of DBIA improves the sharpness and quality of the generated details.

ding applied to distinguish between them effectively. Within the U-Net architecture, specifically in the mid-block where semantic information is most concentrated, we incorporate a task-specific enhancement by upgrading the original attention module to a Detail-Balancing Interactive Attention. This mechanism aligns the attention regions of both the mask and edge streams, facilitating more efficient interaction and complementarity between the two.

**Batch-Discriminative Embedding** Inspired by (Fu et al., 2024; Long et al., 2024), we incorporate additional discriminative labels , *i.e.*, $d_{lab}$, within batches to enable a single stable diffusion model to generate multiple types of outputs simultaneously, ensuring seamless domain processing without mutual interference. Specifically, the $d_{lab}$ is first represented in binary form and then encoded using positional encoding (Mildenhall et al., 2021). After passing through a learnable projection head, the resulting batch-discriminative embeddings are combined with the time embeddings through an element-wise addition. The combined embeddings are then fed into the ResBlocks, enhancing the model's capacity to produce batch-specific outputs. The process is illustrated in Fig. 3

**Detail-Balancing Interactive Attention** To ensure the continuity and alignment of mask and edge, as well as to harmonize their semantic cues and attentioned areas, we introduce Detail-Balancing Interactive Attention (DBIA) , which includes, similar to traditional architectures, a self-attention module and a vanilla cross-attention module, along with our newly designed Mutual Cross-Domain Attention mechanism. DBIA is designed to facilitate a simple yet effective exchange of information between the two domains, with the ultimate goal of generating outputs that are well-aligned in terms of both edge details and semantic content.

Specifically, we initialize the Mutual Cross-Domain Attention module with the parameters of the self-attention module, and project the mask and edge features to produce the query ($\mathbf{Q}_m$), key ($\mathbf{K}_m$), and value ($\mathbf{V}_m$) matrices for the mask, and correspondingly for the edge ($\mathbf{Q}_e$, $\mathbf{K}_e$, $\mathbf{V}_e$), then, we

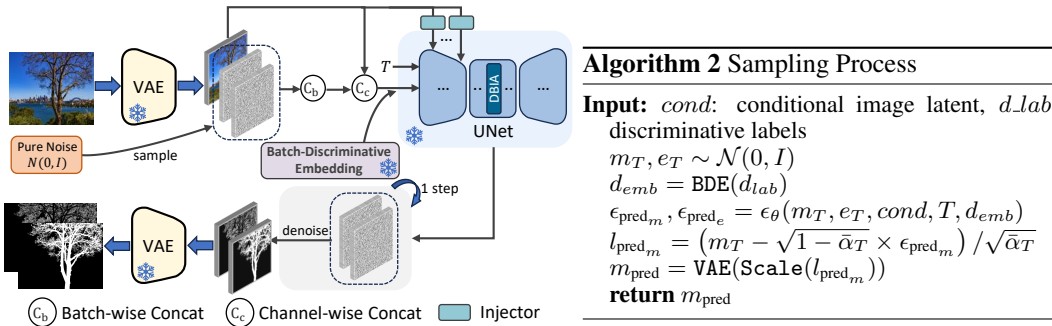

Figure 5: Inference procedure of our DiffDIS.

perform mutual scaled dot-product attention between these two sets, utilizing the query features to effectively query the corresponding key and value features from the alternative domain. The formulation is as follows:

$$\text{Attention}_m = \text{softmax}\left(\frac{\mathbf{Q}_m \mathbf{K}_e^T}{\sqrt{d_k}}\right)\mathbf{V}_e, \qquad \text{Attention}_e = \text{softmax}\left(\frac{\mathbf{Q}_e \mathbf{K}_m^T}{\sqrt{d_k}}\right)\mathbf{V}_m \qquad (6)$$

This method ensures a direct and efficient exchange of information between two domains, fostering a robust interaction that enhances the overall feature representation.

Our method differs from the approach of (Fu et al., 2024) in how we handle feature interaction. In their work, keys ($\mathbf{K}$) and values ($\mathbf{V}$) are concatenated into a unified representation, and cross-attention is computed using domain-specific queries ($\mathbf{Q}$) on this combined feature. Our method performs targeted attention between mask and edge features, enhancing discrimination and alignment while preserving information integrity (Tab. 5). This approach has been observed to not only enhance the fine detail rendering but also, to our delight, significantly reduce the stochastic tendencies inherent in diffusion processes (See in Fig. 4) .

### 4.3 SCALE-WISE CONDITIONAL INJECTION

In traditional diffusion models for dense prediction tasks, conditioning is typically done at the input stage by concatenating along the channel dimension with the variable to be denoised (Ke et al., 2024; Fu et al., 2024). This approach can lead to information loss in later stages. To establish long-range and profound conditional guidance, we introduce Scale-Wise Conditional Injection (SWCI) to enhance multi-granular perception and facilitate deep visual interactions. Specifically, we incorporate multi-scale conditions into the corresponding layers of the U-Net encoder (See Fig. 3) and employ three injector heads, each composed of a simple convolution layer and two zero convolution layers (Zhang et al., 2023) (See Fig. 3). Each injector head receives conditional latent code that is resized to corresponding scales as input. The resulting outputs are then integrated at the junction of the last three layers in the U-Net encoder, facilitating the generation of more authentic structural details. In this way, we introduce multi-granularity information for semantic and structural perception, without causing excessive interference with RGB texture.

### 4.4 ONE-STEP MASK SAMPLING

A depiction of the sampling pipeline can be seen in Fig. 5. During inference, we first encode the RGB image into the latent space using a VAE encoder. Next, we sample the starting variable from standard Gaussian noise, which serves as the initialization for both the mask and edge in the latent space. Similar to the training phase, these two components are concatenated in a batch, with batch-discriminative embedding applied to effectively distinguish between them. The concatenated components, conditioned on the RGB latent representation, are then fed into the U-Net to predict the noise. Building upon the established DDPM approach (Ho et al., 2020), we implement a streamlined one-step sampling process, as detailed in Fig. 5. Finally, the mask and edge map are decoded from the latent code using the VAE decoder and are post-processed by averaging the channels.

## 5 EXPERIMENTS

### 5.1 EXPERIMENTAL SETTINGS

**Datasets and Metrics** Similar to previous works (Yu et al., 2024; Kim et al., 2022), we conducted training on the DIS5K training dataset, which consists of 3,000 images spanning 225 categories. Validation and testing were performed on the DIS5K validation and test datasets, referred to as DIS-VD and DIS-TE, respectively. The DIS-TE dataset is further divided into four subsets (DIS-TE1, DIS-TE2, DIS-TE3, DIS-TE4), each containing 500 images with progressively more complex morphological structures. Following MVANet (Yu et al., 2024), we employ a total of five evaluation metrics, concentrating on measuring the precision of the foreground areas as well as the intricacy of the structural details across the compared models, including max F-measure ($F_\beta^{max}$) (Perazzi et al., 2012), weighted F-measure ($F_\beta^\omega$) (Margolin et al., 2014), structural similarity measure ($S_m$) (Fan et al., 2017), E-measure ($E_\phi^m$) (Fan et al., 2018) and mean absolute error (MAE, $\mathcal{M}$) (Perazzi et al., 2012).

**Implementation Details** Experiments were implemented in PyTorch and conducted on a single NVIDIA H800 GPU. During training, the original images were resized to $1024 \times 1024$ for training. We use SD V2.1 (Rombach et al., 2022) as our backbone, and initialize the model with the parameters from SD-Turbo (Sauer et al., 2023). To mitigate the risk of overfitting in diffusion models when trained on a relatively small dataset, we apply several data augmentation techniques, including random horizontal flipping, cropping, rotation, and CutMix (Yun et al., 2019). For optimization, we use the Adam optimizer, setting the initial learning rate to $3 \times 10^{-5}$. The batch size is configured as 4. The maximum number of training epochs was set to 90. During evaluation, we binarize the predicted maps to filter out minor noise for accuracy calculation.

### 5.2 COMPARISON

**Quantitative Evaluation** In this study, we benchmark our proposed DiffDIS against six DIS-only models, including IS-Net (Qin et al., 2022), FP-DIS (Zhou et al., 2023), UDUN (Pei et al., 2023), InSPyReNet (Kim et al., 2022), BiRefNet (Zheng et al., 2024), and MVANet (Yu et al., 2024). Additionally, we incorporate GenPercept (Xu et al., 2024), a diffusion-based model that has been experimentally applied to DIS. We also include four widely recognized segmentation models: BSANet (Zhu et al., 2022), ISDNet (Guo et al., 2022), IFA (Hu et al., 2022), and PGNet (Xie et al., 2022). For a fair comparison, we standardize the input size of the comparison models to $1024 \times 1024$. The results, as shown in Tab. 2, indicate that our method outperforms others and achieves state-of-the-art performance.

**Qualitative Evaluation** Fig. 6 presents a qualitative comparison of our approach with previous state-of-the-art methods, highlighting our method's enhanced capability to perceive fine regions with greater clarity. As depicted in Fig. 6, our model adeptly captures precise object localization and edge details across a variety of complex scenes, showcasing its robust performance in high-accuracy segmentation tasks.

### 5.3 ABLATION

In this section, we analyze the effects of each component and evaluate the impact of various pre-trained parameters and denoising paradigms on experimental accuracy. All results are tested on the DIS-VD dataset.

**Effectiveness of each component** In Tab. 3, the first row represents the SD model using a single-step denoising paradigm as described in Eqn. 4, initialized with SD-Turbo pre-trained parameters. Subsequent rows represent the incremental addition of auxiliary edge prediction, the DBIA mechanism, the SWCI module, and CutMix data augmentation, respectively. The pro-

Table 3: Ablation experiments of components.

| Edge | DBIA | SWCI | CutMix | $F_\beta^{max} \uparrow$ | $E_\phi^m \uparrow$ | $S_m \uparrow$ | $\mathcal{M}\downarrow$ |
|---|---|---|---|---|---|---|---|
| | | | | 0.882 | 0.930 | 0.880 | 0.039 |
| ✓ | | | | 0.891 | 0.934 | 0.888 | 0.036 |
| ✓ | ✓ | | | 0.896 | 0.943 | 0.895 | 0.032 |
| ✓ | ✓ | ✓ | | 0.897 | 0.945 | 0.899 | 0.030 |
| ✓ | ✓ | ✓ | ✓ | **0.908** | **0.948** | **0.904** | **0.029** |

gressive trend illustrates the utility of each module, demonstrating their collective contribution to the model's performance.

Table 2: Quantitative comparison of DIS5K with 11 representative methods. ↓ represents the lower value is better, while ↑ represents the higher value is better. The best score is highlighted in **bold**, and the second is underlined.

| Datasets | Metric | BSANet | ISDNet | IFA | PGNet | IS-Net | FP-DIS | UDUN | InSPyReNet | BiRefNet | MVANet | GenPercept | Ours |
|---|---|---|---|---|---|---|---|---|---|---|---|---|---|
| *DIS-VD* | $F_\beta^{max}\uparrow$ | 0.738 | 0.763 | 0.749 | 0.798 | 0.791 | 0.823 | 0.823 | 0.889 | 0.897 | 0.904 | 0.844 | **0.908** |
| | $F_\beta^\omega\uparrow$ | 0.615 | 0.691 | 0.653 | 0.733 | 0.717 | 0.763 | 0.763 | 0.834 | 0.863 | 0.863 | 0.824 | **0.888** |
| | $E_\phi^m\uparrow$ | 0.807 | 0.852 | 0.829 | 0.879 | 0.856 | 0.891 | 0.892 | 0.914 | 0.937 | 0.941 | 0.924 | **0.948** |
| | $S_m\uparrow$ | 0.786 | 0.803 | 0.785 | 0.824 | 0.813 | 0.843 | 0.838 | 0.900 | **0.905** | **0.905** | 0.863 | 0.904 |
| | $\mathcal{M}\downarrow$ | 0.100 | 0.080 | 0.088 | 0.067 | 0.074 | 0.062 | 0.059 | 0.042 | 0.036 | 0.034 | 0.044 | **0.029** |
| *DIS-TE1* | $F_\beta^{max}\uparrow$ | 0.683 | 0.717 | 0.673 | 0.754 | 0.740 | 0.784 | 0.784 | 0.845 | 0.866 | 0.873 | 0.807 | **0.883** |
| | $F_\beta^\omega\uparrow$ | 0.545 | 0.643 | 0.573 | 0.680 | 0.662 | 0.713 | 0.720 | 0.788 | 0.829 | 0.823 | 0.781 | **0.862** |
| | $E_\phi^m\uparrow$ | 0.773 | 0.824 | 0.785 | 0.848 | 0.820 | 0.860 | 0.864 | 0.874 | 0.917 | 0.911 | 0.889 | **0.933** |
| | $S_m\uparrow$ | 0.754 | 0.782 | 0.746 | 0.800 | 0.787 | 0.821 | 0.817 | 0.873 | 0.889 | 0.879 | 0.852 | **0.891** |
| | $\mathcal{M}\downarrow$ | 0.098 | 0.077 | 0.088 | 0.067 | 0.074 | 0.060 | 0.059 | 0.043 | 0.036 | 0.037 | 0.043 | **0.030** |
| *DIS-TE2* | $F_\beta^{max}\uparrow$ | 0.752 | 0.783 | 0.758 | 0.807 | 0.799 | 0.827 | 0.829 | 0.894 | 0.906 | 0.916 | 0.849 | **0.917** |
| | $F_\beta^\omega\uparrow$ | 0.628 | 0.714 | 0.666 | 0.743 | 0.728 | 0.767 | 0.768 | 0.846 | 0.876 | 0.874 | 0.827 | **0.895** |
| | $E_\phi^m\uparrow$ | 0.815 | 0.865 | 0.835 | 0.880 | 0.858 | 0.893 | 0.886 | 0.916 | 0.943 | 0.944 | 0.922 | **0.951** |
| | $S_m\uparrow$ | 0.794 | 0.817 | 0.793 | 0.833 | 0.823 | 0.845 | 0.843 | 0.905 | 0.913 | **0.915** | 0.869 | 0.913 |
| | $\mathcal{M}\downarrow$ | 0.098 | 0.072 | 0.085 | 0.065 | 0.070 | 0.059 | 0.058 | 0.036 | 0.031 | 0.030 | 0.042 | **0.026** |
| *DIS-TE3* | $F_\beta^{max}\uparrow$ | 0.783 | 0.817 | 0.797 | 0.843 | 0.830 | 0.868 | 0.865 | 0.919 | 0.920 | 0.929 | 0.862 | **0.934** |
| | $F_\beta^\omega\uparrow$ | 0.660 | 0.747 | 0.705 | 0.785 | 0.758 | 0.811 | 0.809 | 0.871 | 0.888 | 0.890 | 0.839 | **0.916** |
| | $E_\phi^m\uparrow$ | 0.840 | 0.893 | 0.861 | 0.911 | 0.883 | 0.922 | 0.917 | 0.940 | 0.951 | 0.954 | 0.935 | **0.964** |
| | $S_m\uparrow$ | 0.814 | 0.834 | 0.815 | 0.844 | 0.836 | 0.871 | 0.865 | 0.918 | 0.918 | **0.920** | 0.869 | 0.919 |
| | $\mathcal{M}\downarrow$ | 0.090 | 0.065 | 0.077 | 0.056 | 0.064 | 0.049 | 0.050 | 0.034 | 0.029 | 0.031 | 0.042 | **0.025** |
| *DIS-TE4* | $F_\beta^{max}\uparrow$ | 0.757 | 0.794 | 0.790 | 0.831 | 0.827 | 0.846 | 0.846 | 0.905 | 0.906 | **0.912** | 0.841 | 0.909 |
| | $F_\beta^\omega\uparrow$ | 0.640 | 0.725 | 0.700 | 0.774 | 0.753 | 0.788 | 0.792 | 0.848 | 0.866 | 0.857 | 0.823 | **0.893** |
| | $E_\phi^m\uparrow$ | 0.815 | 0.873 | 0.847 | 0.899 | 0.870 | 0.906 | 0.901 | 0.936 | 0.940 | 0.944 | 0.934 | **0.955** |
| | $S_m\uparrow$ | 0.794 | 0.815 | 0.841 | 0.811 | 0.830 | 0.852 | 0.849 | **0.905** | 0.902 | 0.903 | 0.849 | 0.896 |
| | $\mathcal{M}\downarrow$ | 0.107 | 0.079 | 0.085 | 0.065 | 0.072 | 0.061 | 0.059 | 0.042 | 0.038 | 0.041 | 0.049 | **0.032** |
| *Overall DIS-TE(1-4)* | $F_\beta^{max}\uparrow$ | 0.744 | 0.778 | 0.755 | 0.809 | 0.799 | 0.831 | 0.831 | 0.891 | 0.900 | 0.908 | 0.840 | **0.911** |
| | $F_\beta^\omega\uparrow$ | 0.618 | 0.707 | 0.661 | 0.746 | 0.726 | 0.770 | 0.772 | 0.838 | 0.865 | 0.861 | 0.817 | **0.892** |
| | $E_\phi^m\uparrow$ | 0.811 | 0.864 | 0.832 | 0.885 | 0.858 | 0.895 | 0.892 | 0.917 | 0.938 | 0.938 | 0.920 | **0.950** |
| | $S_m\uparrow$ | 0.789 | 0.812 | 0.791 | 0.830 | 0.819 | 0.847 | 0.844 | 0.900 | **0.906** | 0.904 | 0.860 | 0.905 |
| | $\mathcal{M}\downarrow$ | 0.098 | 0.073 | 0.084 | 0.063 | 0.070 | 0.057 | 0.057 | 0.039 | 0.034 | 0.035 | 0.044 | **0.028** |

Table 4: Ablation experiments of the pre-trained parameters and denoising steps. The asterisk (*) indicates that the timestep is not fixed during training.

| Pre-trained Params | Train step | Infer step | $F_\beta^{max}\uparrow$ | $E_\phi^m\uparrow$ | $S_m\uparrow$ | $\mathcal{M}\downarrow$ | Inf. Time ↓ |
|---|---|---|---|---|---|---|---|
| Train from scratch | 1 | 1 | 0.682 | 0.758 | 0.687 | 0.103 | **0.33** |
| SD-Turbo | 1 | 1 | **0.883** | 0.930 | 0.880 | 0.039 | **0.33** |
| | 1* | 2 | 0.876 | 0.918 | 0.878 | 0.041 | 0.52 |
| | 2 | 2 | 0.879 | **0.933** | **0.881** | **0.038** | 0.52 |
| SDV2.1 | 1 | 1 | 0.853 | 0.910 | 0.862 | 0.048 | **0.33** |
| | 1* | 10 | 0.837 | 0.888 | 0.831 | 0.052 | 1.40 |

**Diverse Pre-trained Parameters and Denoising Steps** As shown in Tab. 4, to further investigate the impact of SD's powerful pretrained prior on high-resolution DIS tasks and to explore the effects of various denoising paradigms, we experimented with the following variants based on the vanilla SD backbone. All experiments shared the same network architecture:

By comparing the results of the 1st, 2nd, and 5th rows, which correspond to different pretrained parameters loaded into the U-Net, it's evident that SD-Turbo outperforms others in one-step denoising. This superior performance can be attributed to its efficient, few-step image generation capabilities acquired through distillation. The 2nd, 3rd, and 4th rows of Table 4 illustrate the varying impacts of different few-step denoising paradigms. Our approach, as mentioned in the paper, is reflected in the 2nd row. The 4th row represents a fixed two-step denoising paradigm with timesteps [999, 499] used for both training and testing, which, compared to the second row, shows a slightly better accuracy but at the expense of doubled inference and training time. The 3rd row modifies the 4th by introducing a random timestep selection from (999, 499) during training, followed by a one-step denoising process. During testing, a fixed two-step denoising procedure is employed, using the timesteps [999, 499]. While this approach reduces training time, it results in lower accuracy than the 2nd row. The performance decrease is probably because the model wasn't consistently trained for the specific two-step denoising needed at test time. The last two rows of Table 4 demonstrate the compara-

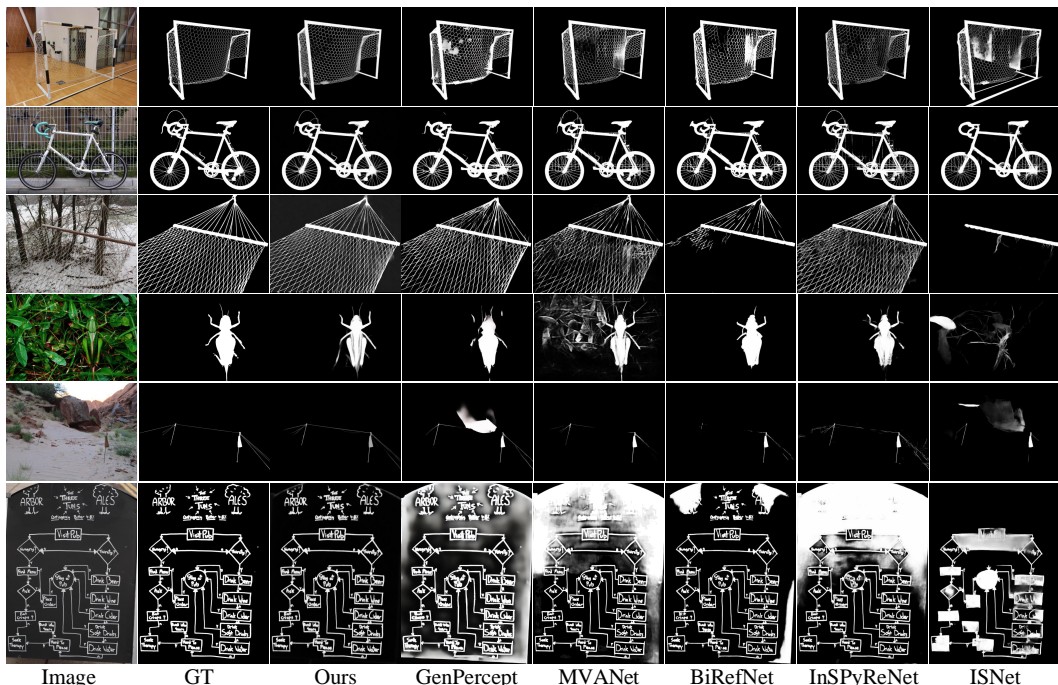

| Image | GT | Ours | GenPercept | MVANet | BiRefNet | InSPyReNet | ISNet |

Figure 6: Visual comparison of different DIS methods.

tive efficacy between one-step denoising paradigms and traditional multi-step approaches (Ke et al., 2024; Rombach et al., 2022) in the context of high-precision, HR object segmentation. The final row, which employs a random timestep between 0 and 1000 to introduce noise into the mask latent during training, followed by a ten-step denoising process during testing, similar to (Ke et al., 2024), yields significantly lower results than one-step paradigms. This diminished performance is attributed to the randomness of multi-step denoising, where stochastic variations are progressively amplified at each step. This leads to increased unpredictability and generative characteristics, conflicting with the deterministic nature of binary segmentation tasks that require unambiguous outputs.

**Comparison of Different Interaction Methods in DBIA** To compare the differences in cross-domain interaction methods between our approach and (Fu et al., 2024), we replaced the interaction method in the additional cross-attention mechanism of DBIA with a fusion-oriented approach similar to that used in (Fu et al., 2024). As shown in the 2$^{nd}$ line, our targeted interaction is more likely to boost generation accuracy and enhance information exchange.

Table 5: Ablation experiments of the interaction in DBIA.

| Edge | DBIA | DBIA$_{geo}$ | $F_\beta^{max} \uparrow$ | $E_\phi^m \uparrow$ | $S_m \uparrow$ | $\mathcal{M} \downarrow$ |
|---|---|---|---|---|---|---|
| ✓ | ✓ | | **0.896** | **0.943** | **0.895** | **0.032** |
| ✓ | | ✓ | 0.891 | 0.940 | 0.894 | 0.035 |

## 6 CONCLUSION

In this paper, we made the attempt to harness the exceptional performance and prior knowledge of diffusion architectures to transcend the limitations of traditional discriminative learning-based frameworks in HR, fine-grained object segmentation, aiming at generating detailed binary maps at high resolutions, while demonstrating impressive accuracy and swift processing. Our approach used a one-step denoising paradigm to generate detailed binary maps quickly and accurately. To handle the complexity and detail richness of DIS segmentation, we introduced additional edge constraints and upgraded the attention module to Detail-Balancing Interactive Attention, enhancing both detail clarity and the generative certainty of the diffusion model. We also incorporated multi-scale conditional injection into the U-Net, introducing multi-granularity information for enhanced semantic and structural perception. Extensive experiments demonstrate DiffDIS's excellent performance on the DIS dataset.

## 7 ACKNOWLEDGEMENT

This work was supported by the National Natural Science Foundation of China under Grant 62431004 and 62276046, and by Dalian Science and Technology Innovation Foundation under Grant 2023JJ12GX015.

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
