# OpenReview forum: "High-Precision Dichotomous Image Segmentation via Probing Diffusion Capacity"
_ICLR.cc/2025/Conference — ICLR 2025 Poster_

### Official Review · Reviewer_8bJM · 2024-10-27

**Soundness:** 3
**Presentation:** 3
**Contribution:** 3
**Rating:** 6
**Confidence:** 4

**Summary:**

This paper proposes a DiffDIS framework based on SD V2.1 to tackle the fine-grained dichotomous image segmentation task. The proposed DiffDIS finetunes the diffusion U-Net in the VAE-encoded latent space and introduces several modifications to enhance the edge perception, including:  1) The edge-assisted training strategy introduces batch-discriminative embedding to enable the mask and edge prediction in a single SD model and conducts interactive attention between the mask and edge branches, 2) Add to zero convolution to enhance the condition injection at different scales. The paper is easy to follow. The proposed method only uses single-step denoising to enhance efficiency and achieves SoTA performance on all the benchmarks.

**Strengths:**

+ The proposed method is technically sound and streamlined, unleashing the dimension-reduction ability of VAE and the representation capacity of diffusion U-Net in the perceptional task.
+ The paper is well-organized and easy to follow.
+ The proposed method achieves superior performance gains on the benchmark datasets.
+ Some experimental observations, such as the restorative capacity of VAE (Tab. 1) and the influence of time-step (Tab. 4) might be valuable to the community.

**Weaknesses:**

+ Some concerns about the technical contribution should be clarified: see Q4 and Q5.
+ Some concerns about the robustness and model efficiency should be addressed: see Q2 and Q6
+ Some other concerns about the methodology should be tackled: see Q1 and Q3

**Questions:**

1. In Algorithm 1, the authors finetune diffusion U-Net by only considering t=T and directly optimizing with $x_0$  instead of noise, where the diffusion UNet seems to degenerate into a common UNet and is inconsistent with SD. Why did the authors not train the diffusion U-Net using the consistent objective with SD and deploy an efficient ODE solver to enable efficient inference?
2. Considering the randomness of the noise sampling process, is the model sensitive to the sampled noise in the inference stage? It is suggested that an analysis of the performance be made using varied noise.
3. Can this performance be improved by sending some text embedding given by the caption model instead of using the empty text embedding?
4. What is the difference between the proposed Detail-Balancing Interactive Attention (Eqn.6) and the common cross-attention deployed between the mask and edge feature?
5.  The multi-scale injector introduces condition signals into blocks. There lacks a comparison with the common condition/signal injection methods, such as the cross-attention in SD.
6. Since VAE contains a large number of parameters, the authors should give a comparison in terms of the inference speed and parameters with state-of-the-art methods.

---

> ### Author Response · Authors · 2024-11-26
>
> >**Q1: Why optimizing with $x_0$ instead of noise \& UNet degeneration \& other approach for efficient inference**
>
> Thank you for raising these valuable concerns, here are our clarifications:
>
> Our one-step denoising paradigm yields the same outcome whether we supervise the noise $\epsilon\_{\theta}(x\_{T}, T, cond, d\_{lab})$ or the denoised $x_0$ derived from it. This is due to the linear relationship between the two, which allows for an easy transformation using the formula provided (See Eqn. 4).
>
> Moreover, we believe that the UNet in our model is not degenerated to a common UNet. Similar to SD, the UNet still predicts the noise distribution. The core of one-step denoising lies in simplifying and streamlining the multi-step denoising paradigm, and thereby minimizing the generative randomness introduced by noise and achieving reliable outcomes in perceptual tasks.
>
> Regarding other efficient inference methods based on ODE solvers, we have indeed employed a consistent objective training approach with a 10-step denoising strategy (See Tab. 4, last row) based on DDIM. However, we found that it did not outperform the direct prediction of noise from $x_T$. We believe this is because multi-step denoising introduces noise variables iteratively, which can lead to results that are more generative in nature, while one-step denoising is more appropriate for binary segmentation tasks that require high determinism.
>
> As for other ODE solvers and inference optimizations, we will further exploring these avenues in our future work.
>
>
> >**Q2: Sensitivity of noise during sampling**
>
> Thank you for your insightful comment. We conducted new experiments with the following setup:
>
> Multiple Noise Predictions: For given samples, we used $5$ distinct noise for prediction. We evaluated the results of each noise-predicted mask individually and also the average result across all $5$ predictions.
>
> Pyramid Noise Initialization: We initialized the input with pyramid noise and compared the resulting predictions to those obtained with our standard noise initialization.
>
>
> |   Metric   | $F_{\beta}^{max} \uparrow$ | $F_{\beta}^{\omega} \uparrow$| $E_{\phi}^m \uparrow$ | $S_m \uparrow$ | $\mathcal{M} \downarrow$ |
> |------|-----|-----|-----|-----|-----|
> |Ours(noise1)| 0.916 |0.885|0.946|0.902|0.030|
> |Ours(noise2)| 0.918 |0.888|0.949|0.905|0.028|
> |Ours(noise3)| 0.917 |0.885|0.947|0.903|0.030|
> |Ours(noise4)| 0.918 |0.888|0.949|0.903|0.029|
> |Ours(noise5)| 0.918 |0.889|0.949|0.905|0.028|
> |Ours(average)| 0.918 |0.889|0.950|0.904|0.029|
> |Ours(pyramid noise)| 0.917 |0.885|0.946|0.902|0.030|
> |Ours| 0.918 |0.888|0.948|0.904|0.029|
>
> As shown in the result above, DiffDIS is stable to the noise sampled during inference due to our streamlined one-step denoising paradigm.
>
>
> >**Q3: Integration of additional text information**
>
> Thank you for your valuable suggestion. We have experimented with CLIP to encode text embeddings for images, but found it didn't lead to any improvements. We believe that this is because the semantic information in CLIP's text embeddings is too broad to provide targeted guidance for segmenting individual objects.
>
> We will continue to explore ways to incorporate additional text cues from other label models to further enhance our model's performance.

---

> ### Author Response · Authors · 2024-11-26
>
> >**Q4: Difference between DBIA and common cross-attention**
>
> Our Detail-Balancing Interactive Attention (DBIA) enhances the standard attention mechanisms in the SD network by introducing a mutual cross-domain attention specifically for edge and mask features, where we employ a commonly-used cross-attention design----queries from one domain interact with the keys and values from the other domain. This mechanism leads to a deep contextual understanding, while enabling better alignment and focus on subtle features. It  ensures that these details are not overlooked and are effectively balanced between the two domains.
>
> >**Q5: Other attempts of scale-wise injection**
>
> Thank you for the suggestion of alternative approaches for the multi-scale injection.
> In response to this issue, we conducted new experiments with the following setup:
>
> We first incorporated condition signals into the network blocks using a combination of simple convolution operations and summation.
> Additionally, following the authors' suggestion, we enabled the scale-wise conditions to interact with the network features via extra cross-attention mechanisms.
> The results are shown in the table below:
> |   Metric   | $F_{\beta}^{max} \uparrow$ | $F_{\beta}^{\omega} \uparrow$| $E_{\phi}^m \uparrow$ | $S_m \uparrow$ | $\mathcal{M} \downarrow$ |
> |------|-----|-----|-----|-----|-----|
> |Ours_SWCI_simp| 0.914 |0.884|0.944|0.901|0.032|
> |Ours_SWCI_cross| 0.918 |0.887|0.946|0.904|0.031|
> |Ours| 0.918 |0.888|0.948|0.904|0.029|
>
>
> >**Q6: Inference speed and parameters**
>
> We measure the inference speed and present a comparison of our FPS and parameters with other methods below (all tested on a single NVIDIA 3090 GPU):
>
> |      | BSANet | ISDNet| IFA | PGNet | IS-Net | FP-DIS | UDUN | InSPyReNet | BiRefNet | MVANet | Geowizard | Ours |
> |------|-----|-----|-----|-----|-----|-----|-----|-----|-----|-----|-----|-----|
> | Parameters (M) | 131.1 | 111.5 | 111.4 | 150.1 | 176.6 | -   | 100.2 | 350 | 844 | 369 | 3460 | 3480 |
> | FPS  | 35   | 78   | 33.7 | 64.3 | 51.3 | -   | 45.5 | 2.2 | 4.1 | 4.6 | 0.95 | 0.58 |
>
> Despite our model's higher parameter count and relatively slower inference speed, it represents an early attempt at using diffusion for high-precision segmentation. As more optimized architectures emerge, such as pruning techniques for SD models, we plan to exploring more efficient solutions in the future to improve our model's performance.

---

> > ### Comment · Reviewer_8bJM · 2024-11-27
> > **The response has addressed my major concerns.**
> >
> > Thanks for the effort of the authors. I think that most of my concerns have been addressed. Hence, I am happy to keep my score.

---

### Official Review · Reviewer_U4V6 · 2024-11-02

**Soundness:** 3
**Presentation:** 3
**Contribution:** 3
**Rating:** 6
**Confidence:** 3

**Summary:**

This paper address the problem of Dichotomous Image Segmentation (DIS) with a generative foundation model, StableDiffusion, by modifying with several key modules.

**Strengths:**

1. The proposed method reaches SOTA performance and beat other concurrent diffusion-based approach on DIS datasets.
2. This is an early attempt that uses a pre-trained generative model for challenging DIS task.
3. The method is efficient in comparison to the line of work that follows SegDiff, which runs diffusion process for more time-steps.
4. The ablation studies include both quantitative numbers and qualitative visualizations, which are helpful for understanding how the whole framework is designed.

**Weaknesses:**

1. The application scenario seems limited. The task setting is only limited to Dichotomous Image Segmentation. It could be more convincing if the authors can also address the applicability of this approach in more settings, e.g. image matting, foreground object segmentation, edge detection.
2. Running diffusion for one-step for segmentation is not a great contribution. This paper might miss some related work that is in the line of DDPM-Seg [1]. A lot of recent work that uses StableDiffusion for unsupervised semantic segmentation and open-vocabulary segmentation is indeed one-step in inference time.
3. The methods are not novel. From the ablation studies, the most prominent modules are Batch-Discriminative Embedding and Detail-Balancing Interactive Attention (DBIA). However, Batch-Discriminative Embedding is proposed by previous work and this work is more of applying that module for DIS task. DBIA is a modified attention module but specifically designed for DIS.

[1] Label-Efficient Semantic Segmentation with Diffusion Models. ICLR 2022.

**Questions:**

1. Latent diffusion models such as SD would map the original image into low-dimensional latents e.g. 32x32x4 for 512x512x3 input. I do not understand well how the low dimensional latent is decoded to high-resolution mask.
2. I would like to see if the proposed methods can be transferrable to other architectures, e.g. pixel-space diffusion UNets and latent DiT.

The followings seem typos or grammar issues, which do not affect my ratings:
1. line 64, "in balance receptive field expansion" should be "in balancing ..."
2. line 72, "It’s power is" should be "Its power is"
3. line 363, "conloution" should be "convolution"
4. line 530, "attmpt" should be "attempt"

---

> ### Author Response · Authors · 2024-11-26
>
> >**W1: Performance on other scenario:**
>
> Thank you for your constructive comments. We conduct extra experiments on HRSOD benchmark. Following[1], our model is trained with the combination of HRSOD-TR and UHRSD-TR, yielding the following results. We will show more applications in the final paper.
>
> |   Metric   | $F_{\beta}^{max} \uparrow$ | $F_{\beta}^{\omega} \uparrow$| $E_{\phi}^m \uparrow$ | $S_m \uparrow$ | $\mathcal{M} \downarrow$ |
> |------|-----|-----|-----|-----|-----|
> |PGNet| 0.937 |0.900|0.958|0.937|0.020|
> |InSPyReNet| 0.956 |0.922|0.951|0.956|0.018|
> |BiRefNet| 0.953 |0.931|0.955|0.956|0.016|
> |Ours| 0.964 |0.959|0.971|0.961|0.010|
>
> [1]Pyramid Grafting Network for One-Stage High Resolution  Saliency Detection. CVPR 2022.
>
> >**W2: Novelty of one-step denoising in DiffDIS:**
>
> Thank you for highlighting this interesting concern. However, we would like to clarify a few points regarding DiffDIS’s relation to DDPM-Seg and other one-step denoising works:
>
>
> Firstly, DDPM-Seg conducts full-step denoising during inference, extracting semantic information from specific U-Net layers at certain reverse steps for segmentation. In contrast, our method starts from a fully noised latent at step 999 and directly  accomplishes one-step denoising, skipping intermediate $x_t$ predictions, thus presenting an end-to-end model.
>
> Regarding other works based on DDPM-Seg that utilize one-step denoising, they typically extract features from frozen SD and use these features as visual /semantic cues in additional adapter/refiner modules. This kind of design often leads to additional complexities in module design and increased computational overhead.
> Moreover, in complex scenes with fine details, straightforward feature extraction and fusion are insufficient for accurate recovery, thereby failing to fully leverage the inherent priors within SD by freezing the model. And, the randomly selected $t$ can also introduce generative randomness into the results.
>
> Our one-step denoising paradigm, as a simplified version of the DDPM reverse process, minimizes the generative randomness introduced by noise, and makes more efficient use of SD's priors, making it more suitable for high-precision binary segmentation tasks that demand high determinism.
>
> We appreciate the opportunity to clarify these points and thank you for raising this issue. It has been a valuable insight, and we will consider how to better utilize intermediate features from the U-Net in future work.
>
> >**W3: Novelty of BDE \& DBIA**
>
> We appreciate the feedback, and would like to clarify that our core contributions extend beyond these modules themselves and the use of BDE and DBIA serves to support the broader goals.
>
> Specifically, our primary focus is on harnessing the power of diffusion models to address the challenges faced by traditional methods in high-precision DIS task, with task-specific designs for higher processing speed, stronger detail perception capability, and higher reliability. This includes the adoption of a one-step denoising strategy and the incorporation of an edge estimation  task to provide enhanced guidance and detail balancing.
>
> To incorporate the edge information to enhance mask generation,
> we design BDE and DBIA,  a suite of necessary, simple yet effective mechanisms, to conduct the essential cross-domain \textbf{distinguishment} and \textbf{communication} between masks and edges, ensuring a seamless and conflict-free domain processing, while significantly enhancing cross-modality information transfer.

---

> > ### Comment · Reviewer_U4V6 · 2024-11-27
> >
> > Thank authors for their efforts and replies to my questions. And thanks for pointing out my misunderstanding about DDPM-Seg and clarifying the difference. Most of my concerns have been addressed. Hence, I would increase my rating to borderline accept.

---

> ### Author Response · Authors · 2024-11-26
>
> >**Q1: How can VAE decode HR image from low-dimension:**
>
> VAEs conceptualize the data distribution as a mixture of several Gaussian distributions. The encoder in a VAE learns to approximate the parameters of this distribution (typically the mean and variance), from which latent variables are sampled, enabling the compression of images into a very low-dimensional space. Additionally, by training the reconstruction capability on a vast array of natural images, VAE develops robust generalization abilities, allowing itself to effectively compress and reconstruct any image. We have demonstrated the superior performance of VAE in specialized tasks, such as binary image reconstruction (See Tab. 1).
>
> For high-resolution segmentation tasks, this indicates that the information are fully preserved within the latent space. All we need to do is to design additional detail-enhancing algorithms to identify and capture these most subtle details, such as edge-assisted estimation, DBIA, and SWCI, aiming for the best denoising effects to refine the representations and enhance the clarity of the features.
>
>
> >**Q2: Transferability to other architectures:**
>
> We appreciate your suggestion to explore the transferability of our methods to other architectures such as pixel-space diffusion UNets and latent DiT.
>
> However, we believe that applying pixel-level diffusion directly to high-precision segmentation tasks is not practical for several reasons:
> Firstly, as input resolutions increase (e.g., 1024x1024), pixel-space diffusion becomes computationally intensive and memory-hungry compared to latent space diffusion. This can be particularly challenging for HR images where the data volume is already substantial.
> Secondly, pixel-space diffusion can significantly slow down processing speeds, exacerbating the already slow inference times for high-resolution images.
> Thirdly, the continuous Gaussian distribution in the VAE latent space is more aligned with the continuous feature space inherent in diffusion processes.
> Lastly, adopting pixel-space diffusion might necessitate training from scratch, bypassing the powerful priors provided by models like SD. This could lead to slower convergence and reduced accuracy, as evidenced by Tab. 4.
>
> Regarding the use of latent DiT as our backbone, we apologize for not having had sufficient time to explore it in the rebuttal period, however, we will investigate it in the final paper.
>
> >**Typo:**
>
> Thank you for pointing out these typos and grammar issues, we appreciate your detailed attention! We will address these issues in our revision and conduct a thorough self-check for any other errors throughout the manuscript. The corrections have been incorporated into the updated PDF file of our revision.

---

### Official Review · Reviewer_V4Fe · 2024-11-04

**Soundness:** 4
**Presentation:** 4
**Contribution:** 4
**Rating:** 6
**Confidence:** 4

**Summary:**

The authors proposed a method for dichotomous segmentation using a Stable Diffusion prior, finding that introducing edges into the segmentation task can enhance performance. They introduced the BDE and DBIA modules, which can distinguish between different tasks and achieve better detail generation capabilities. The method efficiently utilizes one-step sampling and shows significant improvement over previous methods across multiple test projects.

**Strengths:**

1. The author discovered that the introduction of edges can enhance the detail and performance of segmentation. They used batch discriminative embedding to distinguish between edges and segmentation. This is a novel method.
2. The author provided detailed experiments that demonstrate the method's strong performance across multiple aspects, and also included an ablation study to prove the effectiveness of each module.

**Weaknesses:**

1. The description of the one step inference is not comprehensive enough, please see Q2
2. For dichotomous segmentation, using an RGB 3-channel VAE to encode a single-channel segmentation mask might be a bit overkill. As an advancement in dichotomous segmentation, some earlier works have used diffusion models for matting, which also achieved very good results. However, considering that it can produce decent results in just one step, it is acceptable.

**Questions:**

1. In emu-edit, they introduced a method called task embedding, it is a learnable embedding added to the time step embedding, to distinguish the different task in multi-task training. As your Batch-Discriminative Embedding contains more detailed design, do you have any performance comparison between your Batch-Discriminative Embedding and the task embedding?
2. For the one step mask sampling, in 4.4 and the fig 5 only state that it is building upon the established DDPM, but no more detailed description and implemented the one step sampling, could you please provide more details on it?
3. From the paper, it shows that the author used SD2.1 and SD turbo as initialization weights, and there is a channel-wise concat operation before feeding into the UNet, which changes the number of channels in the input layer. I would like to know how the pretrained weights are handled for the input layer when they are used (eg. duplicate/zero).

---

> ### Author Response · Authors · 2024-11-24
>
> >**W2: Overkillness of VAE:**
>
> We thank the reviewer for pointing out this issue! If a single-channel VAE is used, it would require retraining to achieve reconstruction performance comparable to that of the three-channel structure. In our work, we initially aimed to leverage the prior from the SD model, which led us to adopt the classical three-channel VAE it provides. However, the suggestion of examining the potential overkill with fewer channels in VAE is very insightful, and we plan to further investigate this approach in future work.
>
> >**Q1: BDE vs. learnable task embedding in EMU-edit**
>
> Thank you for your suggestion. Our BDE diverges from EMU-edit’s learnable task embedding in several key aspects. We employ one-dimensional orthogonal vectors, contrasting with the task vectors they fetch from an embedding table. Our embeddings interact solely within the ResBlock layers, whereas theirs are integrated into both residual and cross-attention layers. Given that EMU-edit's source code is not publicly available, we independently implemented learnable task embeddings in DiffDIS, yielding the following results:
>
> |   Metric   | $F_{\beta}^{max} \uparrow$ | $F_{\beta}^{\omega} \uparrow$| $E_{\phi}^m \uparrow$ | $S_m \uparrow$ | $\mathcal{M} \downarrow$ |
> |------|-----|-----|-----|-----|-----|
> |Ours(task embedding)| 0.916 |0.887|0.948|0.905|0.032|
> |Ours(BDE)| 0.918 |0.888|0.948|0.904|0.029|
>
>
> >**Q2: Details on one-step sampling:**
>
> During the inference phase, the one-step denoising process aligns with the training phase by replacing the noised mask and edge latents with Gaussian noise. Using the image latent of the target segmentation as a condition, it guides the generation of noise
> $\epsilon\_{\text{pred}\_m}$. Denoising is then conducted using the process outlined in Fig. 5, where $l\_{\text{pred}\_m} = \left(m\_{T} - \sqrt{1 - \bar{\alpha}\_T} \times \epsilon\_{\text{pred}\_m} \right) / \sqrt{\bar{\alpha}\_T} $. This constitutes the one-step sampling procedure.
>
> >**Q3: How the pretrained weights are handled when doubling input channels:**
>
> For the input convolution layer, we duplicated the 4-channel weights and then scaled all of them by 0.5

---

> > ### Comment · Reviewer_V4Fe · 2024-11-25
> >
> > The author's response solved all my questions, I'd like to keep the score.

---

### Official Review · Reviewer_N1et · 2024-11-04

**Soundness:** 3
**Presentation:** 2
**Contribution:** 2
**Rating:** 6
**Confidence:** 3

**Summary:**

This paper proposes a model named DiffDIS for high-resolution binary image segmentation, incorporating strategies like single-step denoising, edge-assisted generation, and multi-scale conditional injection to enhance segmentation accuracy and inference speed. The authors validate DiffDIS’s performance on the DIS5K dataset, showing promising results. While the design is sound and experimental results are clearly presented, the paper’s novelty and certain implementation details could benefit from further clarification. In the rebuttal, the author addressed these concerns.

**Strengths:**

1.	Multi-level Design Innovations: The paper combines single-step denoising, edge-assisted generation, and multi-scale conditional injection to address the challenges of high-resolution segmentation, balancing speed and detail retention effectively.
2.	Comprehensive Experiments: The experimental setup on the DIS5K dataset is thorough, with comparisons to multiple specialized and general segmentation models. Ablation studies illustrate each component’s contribution, supporting the rationale behind the model design.
3.	Clarity: The paper is well-organized, with clear descriptions of the model and results, making it accessible to readers.

**Weaknesses:**

1.	Clarify Novelty of the Single-Step Denoising: While the single-step denoising strategy indeed boosts inference efficiency, a similar concept has been explored in models like GenPercept. I suggest that the authors clarify if DiffDIS’s single-step denoising incorporates task-specific optimizations for DIS tasks, to better highlight its originality.
	2.	Elaborate on the Edge-Assisted Generation’s Distinctiveness and Adaptation for High-Resolution Segmentation: The edge-assisted generation approach in DiffDIS appears similar to the edge-guided inpainting technique used in EdgeConnect, albeit applied in segmentation rather than inpainting. To avoid the impression that this is a simple adaptation from inpainting, I suggest the authors discuss any specific adjustments or optimizations made for high-resolution segmentation in DiffDIS.
	3.	Advantages of Joint Edge and Mask Prediction with Experimental Validation: DiffDIS performs joint edge and mask prediction, unlike stage-wise processing. Further discussion on the specific advantages of joint prediction, especially in handling fine details and complex boundaries, would strengthen this choice. Additionally, including experimental comparisons between joint prediction and stage-wise prediction would provide valuable insights into its effectiveness.
	4.	Comparative Analysis of Training Time: While DiffDIS’s inference time is shown to be efficient, comparative analysis of training times is absent. Including training time comparisons would provide a more holistic view of DiffDIS’s computational efficiency.

**Questions:**

please see the weakness part

---

> ### Author Response · Authors · 2024-11-24
>
> >**W1: Key differences from GenPercept \& task-specific optimizations in DiffDIS:**
>
> Thank you for your feedback! GenPercept utilizes a deterministic framework that maps conditions (e.g. image latents), directly to the target perceptual output **without undergoing a noise-corruption and denoising process**, while DiffDIS maintains the ability of diffusion models to **extract intricate details from noise by rooting in DDPM**, and employing a simplified denoising paradigm that recovers the target from globally noised variables, which minimizes the randomness inherent in generative models while preserving the detail-extraction capabilities of diffusion models. This design aligns well with high-precision tasks like DIS, leading to superior performance against GenPercept (See Tab. 2).
>
>
> > **W2: Key differences from EdgeConnect:**
>
> Thank you for your concern, however, we beg to differ on this issue:
>
> EdgeConnect used a **two-stage, sequential** prediction strategy, generating the edges of the image to be inpainted first, followed by the image itself. In EdgeConnect, the edges act as essential **intermediates/connectors** between the edge estimation task and image inpainting task  (hence "EdgeConnect" is named).
>
> Different from them, we use a **parallel, interactive** approach that generates edges and masks simultaneously (with BDE helping manage different modalities within each batch), where the edge estimation acts as an **auxiliary task**, aiming at enhancing the mask generation by imposing finer constraints, and diminishing the stochastic nature of diffusion processes.
>
> DiffDIS is distinctly tailored for the DIS task, with a clear focus on achieving **higher inference speed, stronger detail extraction, and higher determinism**, rather than "albeit applied in segmentation instead of inpainting".
>
> > **W3: Comparision of joint predicton \& stage-wise processing:**
>
> In response to your suggestion, we implemented a two-stage mask generation process following EdgeConnect's training workflow :
>
> In the first stage, an edge generation model produces edges as intermediate outputs, which are then used as additional conditions for the mask generation task in the second stage. To ensure fairness, we also integrated the DBIA module in the second stage. The results, along with inference times, are shown below:
>
> |   Metric   | $F_{\beta}^{max} \uparrow$ | $F_{\beta}^{\omega} \uparrow$| $E_{\phi}^m \uparrow$ | $S_m \uparrow$ | $\mathcal{M} \downarrow$ | Inf. Time(s)|
> |------|-----|-----|-----|-----|-----|-----|
> |Stage-wise| 0.896 |0.874|0.941|0.889|0.034|0.59|
> |Ours| 0.918 |0.888|0.948|0.904|0.029|0.33|
>
> As shown in the result above, our joint prediction strategy enhances the network's ability to capture fine details while being more efficient and resource-friendly.
>
> > **W4: Training time comparison:**
>
> Thank you for your suggestion. We compiled a comparison of training times with 11 other methods, and the results are as follows:
>
> |      | BSANet | ISDNet| IFA | PGNet | IS-Net | FP-DIS | UDUN | InSPyReNet | BiRefNet | MVANet | Geowizard | Ours |
> |------|-----|-----|-----|-----|-----|-----|-----|-----|-----|-----|-----|-----|
> | Trainig time (h)  | -    | -    | -    | 24   | -    | -    | -    | -   | -    | 53 |   -   |   75   |

---

### Meta-Review · Area_Chair_YQwM · 2024-12-22

**Metareview:**

While utilizing diffusion model for image segmentation is not new, this paper proposes a variant of diffusion model for high-precision dichotomous image segmentation. During inference, the method is efficient with single-step denoising from random Gaussian noise. Edge information is integrated with mask via Batch Discriminative Embedding (Fu. et, al. 2024).
Many recent or concurrent works on diffusion models for image segmentation rely on single-step denoising. The method here is slightly different as it denoises random Gaussian noise for edge and mask at the last step conditioned on an image. It is not clear though why single step suffices for denoising of mask and edge map, though this can be due to the simplicity of such modalities compared to information-rich images. The Batch Discriminative Embedding block is also not new.
The technical novelty of this paper is not significant, but the method is technically sound, and the results are impressive and convincing. Hence, the AC recommends acceptance of this paper as a poster.

**Additional Comments On Reviewer Discussion:**

Reviewer U4V6 and N1et are both concerned about the novelty of the single-step denoising for image segmentation.
The authors have clarified the difference from other work that uses single-step denoising.

Reviewer 8bJM was skeptical about the method being sensitive to initial noise, as it only involves one denoising step.
Empirical evidence of robustness is given in the rebuttal.

---

### Decision · Program_Chairs · 2025-01-22

Accept (Poster)